# Neoadjuvant and Adjuvant Chemotherapy for Pancreatic Adenocarcinoma: Literature Review and Our Experience of NAC-GS

**DOI:** 10.3390/cancers16050910

**Published:** 2024-02-23

**Authors:** Taku Aoki, Shozo Mori, Keiichi Kubota

**Affiliations:** Department of Hepato-Biliary-Pancreatic Surgery, Dokkyo Medical University, Mibu 321-0293, Tochigi, Japan; shozomori@hotmail.co.jp (S.M.); kubotak@dokkyomed.ac.jp (K.K.)

**Keywords:** pancreatic ductal adenocarcinoma, neoadjuvant, adjuvant, gemcitabine, S-1

## Abstract

**Simple Summary:**

Recent observations have led to an expansion of the role of neoadjuvant treatment (NAT) as a component of a multidisciplinary approach to the treatment of pancreatic ductal adenocarcinoma (PDAC). However, some issues related to this treatment remain unclear, including (1) the appropriate indications for NAT (as opposed to up-front surgery); (2) predictors of response to NAT; (3) the effect of the response to NAT on the efficacy of postoperative adjuvant chemotherapy (AC); and (4) the establishment of an adjuvant treatment policy based on post-neoadjuvant therapy/surgery histopathological findings. In this article, we discuss these issues based on a review of the literature and the authors’ own experience of NAT using gemcitabine plus S-1.

**Abstract:**

In addition to established evidence of the efficacy of adjuvant chemotherapy (AC) for pancreatic ductal adenocarcinoma (PDAC), evidence of the effects of neoadjuvant treatments (NATs), including chemotherapy and chemoradiotherapy, has also been accumulating. Recent results from prospective studies and meta-analyses suggest that NATs may be beneficial not only for borderline resectable PDAC, but also for resectable PDAC, by increasing the likelihood of successful R0 resection, decreasing the likelihood of the development of lymph node metastasis, and improving recurrence-free and overall survival. In addition, response to NAT may be informative for predicting the clinical course after preoperative NAT followed by surgery; in this way, the postoperative treatment strategy can be revised based on the effect of NAT and the post-neoadjuvant therapy/surgery histopathological findings. On the other hand, the response to NAT and AC is also influenced by the tumor biology and the patient’s immune/nutritional status; therefore, planning of the treatment strategy and meticulous management of NAT, surgery, and AC is required on a patient-by-patient basis. Our experience of using gemcitabine plus S-1 showed that this NAT regimen achieved tumor shrinkage and decreased the levels of tumor markers but failed to provide a survival benefit. Our results also suggested that response/adverse events to NAT may be predictive of the efficacy of AC, as well as survival outcomes.

## 1. Introduction

Pancreatic ductal adenocarcinoma (PDAC), the incidence of which appears to be increasing worldwide, is associated with a high mortality rate due to its aggressive biological behavior [1]. PDAC was rated as the third leading cause of cancer-related mortality in the USA in 2023 [2]. In Japan, about 39,000 deaths were attributed to PDAC in 2022, and pancreatic cancer was estimated to be the fourth most common cause of cancer death in men and the third in women [3]. While surgical resection is the only hope for cure in patients with PDAC, the resection rate has remained at approximately 20% in recent years [4,5]. In addition, surgery, as the lone treatment strategy, has reportedly been associated with a high recurrence rate and early cancer death. As such, multidisciplinary approaches, including R0 surgical resection combined with neoadjuvant and adjuvant therapies, have been proposed.

With recent advances in chemotherapeutic and chemoradiotherapeutic regimens, evidence of the benefits of neoadjuvant/adjuvant therapies using these novel regimens has been accumulating. Recent prospective studies have suggested that all patients with resectable or borderline resectable (BR) PDAC are candidates for adjuvant treatments (ATs) [6,7]. However, AT cannot be applied in all patients undergoing surgical resection, mainly due to postoperative complications and/or the frailty of patients, or early recurrence. In addition, dose reduction during AT is frequently needed in postoperative patients, resulting in a weakened efficacy of the treatment. From this standpoint, neoadjuvant therapies (NATs) might be more reasonable, because NATs can be applied to more patients scheduled for curative resections, with more reliable treatment intensity [8]. On the other hand, complications associated with NATs may also preclude curative resection in potentially resectable patients, and, at present, there are no measures to predict the benefits/disadvantages of NATs. A number of excellent reviews and meta-analyses have traced the development of NAT/AT for PDAC and discussed its significance [8,9,10,11]. Herein, we discuss some unresolved issues associated with NAT/AT for PDAC. We also provide a review of our experience with neoadjuvant chemotherapy (NAC) using gemcitabine plus S-1 for PDAC.

## 2. Evidence of the Benefits of Adjuvant/Neoadjuvant Chemotherapy for Pancreatic Adenocarcinoma

Evidence for the benefit of adjuvant chemotherapy (AC) for PDAC is well established, and virtually all patients with resectable and BR disease are considered as candidates for AC [6,7]. Various regimens have been utilized for AC, including gemcitabine monotherapy, S-1 monotherapy, combined therapy with gemcitabine plus capecitabine, and modified FOLFIRINOX [12,13,14,15]; evidence of the benefit of gemcitabine plus nab-paclitaxel is, however, still limited [16]. In the SWOG-S1505 study, although the survival outcomes of patients who received adjuvant gemcitabine plus nab-paclitaxel therapy were similar to those of patients who received adjuvant mFOLFIRINOX therapy, the primary endpoints were not met statistically. Therefore, the National Comprehensive Cancer Network (NCCN) guidelines and the European Society for Medical Oncology (ESMO) Clinical Practice Guidelines recommend modified FOLFIRINOX or gemcitabine plus capecitabine as the preferred regimens for AC for patients with PDAC [6,17]. Meanwhile, in Japan, S-1 monotherapy is the mainstay of AC for PDAC [13], mainly due to concerns about the adverse events associated with more powerful and toxic regimens. Regarding AC, completion of the scheduled regimen (usually lasting six months) with a maintained dose intensity is considered to be of critical importance to prevent recurrence and prolong patient survival. Therefore, the selection of the regimen should be patient-based, and meticulous management of the patient during AC is important.

On the other hand, the significance of adjuvant chemoradiotherapy (CRT) has not yet been established. The ESPAC-1 and EORTIC trial 40,891 showed no additional benefit of adjuvant CRT in patients with PDAC [18]; however, in one retrospective cohort study, some patients benefited more from adjuvant CRT than from adjuvant CT [19].

It should be noted that in an AT setting, the response to or effectiveness of the selected regimens cannot be monitored or evaluated, as the tumor has already been removed. In other words, many patients who undergo up-front resection may receive ineffective ATs with significant toxicity without biological information about the tumor. It is from this viewpoint that the concept of neoadjuvant therapy was conceived, as the effect of the selected chemotherapy regimens can be evaluated, at least in part, by the radiological or biological responses.

The necessity of NAT in patients with BR disease has come to be well recognized, and several clinical guidelines recommend NAT for patients with BR PDAC [6,7,17]. Recent meta-analyses have shown the survival benefit of NAT (NAC or NACRT) for BR PDAC [20,21], while any additional benefit of one over the other of NACRT versus NAC remains unclear. In a retrospective analysis of 884 Japanese patients with BR PDAC, NACRT was associated with a lower resection rate, but also a lower rate of lymph node metastasis and lower rate of local recurrence; however, the overall survival was comparable between the NACRT and NAC groups [22].

Recently, the application of NAC has been expanded to resectable PDAC [11,23]. The Prep-02/JSAP05 study showed that NAC using gemcitabine plus S-1 provided significant survival benefit, while the PREOPANC study showed that NACRT using gemcitabine-based NAC and radiotherapy was associated with an improved median survival period [24,25]. Unno et al. conducted a randomized controlled trial of Neoadjuvant Chemotherapy Using Gemcitabine Plus S-1 (NAC-GS) in 362 patients with resectable or BR PDAC with portal vein involvement. Those authors demonstrated a significantly higher median OS in the NAC-GS arm than in the up-front surgery arm and concluded that NAC-GS could be a new standard treatment strategy for potentially resectable PDAC [24]. The PREOPANIC trial assigned patients with resectable or BR PDAC in a randomized manner to treatment with adjuvant chemotherapy (control arm) or perioperative NACRT (test arm). The patients in the control arm received up-front surgery followed by adjuvant gemcitabine-based chemotherapy, while the patients in the test arm received neoadjuvant gemcitabine-based conformal radiation therapy followed by surgery, followed again by another four months of adjuvant gemcitabine-based chemotherapy. The five-year overall survival rates were 20.5% in the test arm and 6.5% in the control arm [25]. However, the results of meta-analyses have been conflicting [8,11]. More recently, the effectiveness of modified FOLFIRINOX in a neoadjuvant setting has been examined [26]. Based on these observations, the current NCCN guidelines recommend up-front surgery followed by AT for patients with resectable PDAC, but they also advise considering NAT in PDAC patients with high-risk features, e.g., equivocal or indeterminate imaging findings, markedly elevated serum CA19-9, large primary tumors, large regional lymph nodes, excessive weight loss, and extreme pain. Recently, BR PDAC has been re-defined using anatomical (A), biological (B), and conditional (C) factors [27]. Biological factors include elevated serum CA19-9 levels, i.e., exceeding 500 U/mL, and/or regional lymph node metastasis, as diagnosed by biopsy or PET-CT. Based on these circumstances, PDAC patients with elevated serum CA19-9 levels are, even if their tumors are anatomically resectable, candidates for NAT and re-assessment of the expected outcome prior to surgery.

Monitoring of the trends of tumor markers, especially the serum levels of CA19-9, is of great importance in patients receiving NAT. The monitoring of serological markers is simple and provides important information regarding the tumor biology. It has been reported that normalization of serum CA19-9 is not achieved in 30% of patients undergoing up-front surgery, and that the beneficial effect of surgery was reduced in these patients. Another retrospective study showed that patients with a normalized serum CA19-9 level during NAT showed better long-term survival than those with persistently elevated serum CA19-9 levels [28]. Thus, normalizing of the CA19-9 level during NAT could be a potential and ideal goal in patients receiving NAT in order to extract the maximum benefit from surgery and to improve survival outcomes. In cases of CA19-9 non-producing tumors, serum DUPAN-2 can be an alternative target for monitoring.

## 3. Association between the Responses to Neoadjuvant and Adjuvant Therapies

Whether patients should receive AC following curative resection after NAT remains under debate. Another issue that remains controversial is whether the effect of NAT and/or the post-neoadjuvant histopathological findings can provide additional information which could be used when selecting appropriate treatment regimens for AC or not. Reports have been conflicting; currently, it may be a common understanding that patients with pathologically proven node-positive disease or larger tumor sizes are candidates for AC [29,30]. The current NCCN guidelines recommend that in patients who have undergone surgery following prior neoadjuvant chemotherapy, additional postoperative chemotherapy and/or chemoradiation should be considered in those with a positive-margin or R1 resection [7].

As six months of adjuvant chemotherapy is a well-established strategy for PDAC, it would be reasonable to assign some part of the total duration of chemotherapy to the neoadjuvant setting and the rest to the postoperative adjuvant setting. In such cases, the response to NAT, as determined in the post-NAT histopathological findings, could help determine the indications and recommended regimens for AC. When the histopathological response is significant, it would be reasonable to resume the same chemotherapy as the NAT regimen for the remainder of the six months. On the other hand, when the response is minimal, alternative AC regimens may be recommended. The NCCN guidelines also state that the adjuvant therapy options are dependent on the response to the neoadjuvant therapy and other clinical considerations, and that the total duration of systemic therapy is typically six months [7]. In Western countries, FOLFIRINOX or modified FOLFIRINOX is widely used as the regimen for AC; recently, intensive regimens including FOLFIRINOX or gemcitabine plus nab-paclitaxel have been introduced in the NAC setting for resectable disease. As the regimens and potencies of NAT advance, the overall treatment strategies for PDAC will inevitably change.

## 4. Experience at Dokkyo Medical University

### 4.1. Results of Neoadjuvant Chemotherapy Using Gemcitabine Plus S-1

We applied neoadjuvant chemotherapy using gemcitabine plus S-1 (NAC-GS) for all patients with PDAC scheduled to undergo surgery with curative intent between December 2013 and December 2019, with the approval of the local ethical committee of Dokkyo Medical University (Review number: R-27-14J; Study Number: UMIN00041189). We selected this regimen based on the encouraging results of a prospective multi-institutional phase 2 trial by Motoi et al. [31]. The exclusion criteria were patients who refused enrollment in the study and patients who could not take S-1 due to gastric/duodenal stricture caused by the tumor.

The NAC-GS regimen has been described previously [32]. In brief, gemcitabine is given at the dose of 1000 mg/m^2^ on days 1 and 8 of each course. S-1 is administered orally at a dose of 40, 50, or 60 mg/m^2^ twice daily, according to the body surface area (<1.25 m^2^, 1.25–1.5 m^2^, or >1.5 m^2^) for the first 14 consecutive days, followed by a 7-day rest period. Each course is repeated every 21 days. Our patients received two courses of neoadjuvant NAC-GS.

In the Classification of Pancreatic Carcinoma published by the Japan Pancreas Society, BR-PDAC is subclassified into BR-PV disease (SMV/PV invasion only) and BR-A disease (arterial invasion) [33]. BR-PV is defined as follows: No evidence of contact with or invasion of the superior mesenteric artery (SMA), celiac axis (CA), or the common hepatic artery (CHA), but tumor contact with or invasion of the superior mesenteric vein (SMV)/portal vein (PV) of 180 degrees or greater or occlusion of the SMV/PV, not exceeding the inferior border of the duodenum. BR-A is defined as tumor contact with or invasion of the SMA and/or CA of less than 180 degrees, with no stenosis or deformity, or tumor contact or invasion of the CHA without tumor contact or invasion of the proper hepatic artery (PHA) and/or the CA. In our analyses, we divided our patients into three subgroups according to the above classification.

During the study period, 95 patients (65 patients with resectable disease, 20 patients with BR-PV disease, and 7 patients with BR-A disease) received NAC-GS. The study cohort included 48 males and 47 females, with a median age of 69 years. The impact of NAC-GS was evaluated in terms of the tumor size, as determined using enhanced CT scans, and levels of the tumor markers (serum CEA [34], CA19-9, DUPAN-2 [35], Span-1 [36], and ealastase-1 [37]) (Table 1). NAC-GS resulted in a significant decrease in tumor size (pre-NAC GS: median 24.6 mm, range 2.4–70.0 mm; post-NAC GS: median 18.2 mm, range 1.4–64.0 mm, *p* < 0.0001), and the median post/pre-NAC GS tumor size ratio was 0.82. The reduction ratio was similar among the resectable, BR-PV, and BR-A groups (*p* = 0.80). In addition, NAC-GS also resulted in significant decreases in serum levels of CA19-9, DUPAN-2, Span-1, and elastase-1 (pre-NAC GS CA19-9: median 203.0 U/mL, range 2–12,000 U/mL, post-NAC GS CA19-9: median 84.5 U/mL, range 2–12,000 U/mL, median reduction ratio: 0.63, *p* < 0.0001; pre-NAC GS DUPAN-2: median 220.0 U/mL, range 25–14,000 U/mL, post-NAC GS DUPAN-2: median 97.0 U/mL, range 4–3400 U/mL, median reduction ratio: 0.66, *p* < 0.0001; pre-NAC GS Span-1: median 52.0 U/mL, range 1–1600 U/mL, post-NAC GS Span-1: median 25.0 U/mL, range 1–1100 U/mL, median reduction ratio: 0.68, *p* < 0.0001; pre-NAC GS elastase-1: median 218.0 ng/dL, range 25–8620 ng/dL, post-NAC GS elastase-1: median 150.0 ng/dL, range 40–3613 ng/dL, median reduction ratio: 0.61, *p* < 0.0001). On the other hand, the level of CEA was similar before and after NAC-GS therapy (*p* = 0.72). The reduction ratios of CA19-9, DUPAN-2, Span-1, and elastase-1 were similar among the resectable, BR-PV, and BR-A groups.

Next, we compared the resection rate, R0 resection rate, and rate of administration of postoperative adjuvant therapy in the NAC-GS group as compared with the historical control group with resectable, BR-PV, and BR-A PDAC that received up-front surgery in our department (*n* = 104). The results revealed that NAC-GS was associated with an increased R0 resection rate in the patients with BR-PV disease (*p* = 0.03) and an increased rate of administration of adjuvant chemotherapy in patients with resectable disease (*p* = 0.01) (Table 2). Furthermore, the results were comparable between the NAC-GS group and the up-front surgery group.

The five-year overall survival rate was the best in patients with resectable disease (28.1%), followed by patients with BR-PV disease (19.3%), and the worst in patients with BR-A disease (0%). NAC-GS failed to improve the overall survival rate (*p* = 0.37) (Figure 1). The results of subgroup analyses were similar (resectable disease: *p* = 0.38; BR disease: *p* = 0.49). On the other hand, the overall survival rate tended to be better in patients who also received adjuvant chemotherapy (*p* = 0.06) (Figure 2).

Our experience showed that two cycles of NAC-GS induced tumor shrinkage and a decrease in the serum levels of the tumor markers and was also associated with an increased R0 resection rate among patients with BR-PV disease; however, this NAC regimen failed to improve the survival outcomes. Our results suggest that a more powerful NAC regimen might be more successful; therefore, since January 2020, we have introduced an updated NAC regimen using gemcitabine plus nab-paclitaxel in patients with BR-PV disease and added radiotherapy in patients with BR-A disease (Registration ID: UMIN000041189).

### 4.2. Factors Predictive of the Response to NAC-GS

In patients who underwent resection after NAC-GS (*n* = 81), the response to the preoperative therapy was evaluated according to the Evans Classification. The results have been reported previously [38]. In brief, the responses were classified into Evans Grade I (<10% tumor cell destruction) in 19 (23.5%) patients, Evans Grade IIa (10–50% tumor cell destruction) in 49 (60.5%) patients, Evans Grade IIb (51–90% tumor cell destruction) in 11 (13.6%) patients, and Evans Grade III (<10% viable-appearing tumor cells) in 2 (2.4%) patients. No significant differences were found between patients showing Evans Grade I and Grade II/III responses in terms of the relative dose intensity of the NAC-GS, incidence of severe adverse events of NAC-GS, or rate of administration of AC.

The operation time was significantly longer in patients who showed Evans Grade I response to NAC-GS versus Evans Grade II/III (*p* = 0.016). The survival outcomes were unfavorable in patients showing Evans Grade I response as compared with those who showed Evans Grade II/III response to NAC-GS (*p* < 0.001). The one- and three-year overall survival rates in patients who showed Evans Grade I and Evans Grade II/III responses to NAC-GS were 56.7% and 17.6%, and 76.7% and 50.2%, respectively (*p* = 0.001). The one- and three-year relapse-free survival rates in patients who showed Evans Grade I and Evans Grade II/III responses to NAC-GS were 23.4% and 0%, and 57.9% and 34.1%, respectively (*p* = 0.001). Pre-treatment factors that were predictive of an Evans Grade I response were a serum CEA level of >3.6 ng/mL and serum C-reactive protein to albumin ratio of >0.062. In other words, no other tumor markers were significantly related to the response to NAC-GS [38].

### 4.3. Impact of NAC-GS on the Outcomes of Adjuvant Chemotherapy

As mentioned above, our retrospective results showed that NAC-GS exerted no significant influence on the survival outcomes in patients with PDAC, while administration of AC after surgery following NAC-GS was associated with a tendency toward better survival outcomes. We use gemcitabine monotherapy or S-1 monotherapy as adjuvant chemotherapy under these circumstances. In our previous investigation, we estimated the impact of adverse events (AEs) emerging during AC on clinical outcomes in patients who received up-front surgery (*n* = 72) and patients who underwent surgery following NAC-GS (*n* = 77) [39]. The results showed that the development of grade 3/4 AEs during AC were associated with a lower relative dose intensity, lower completion rate of AC, and unfavorable survival outcomes in patients who underwent surgery after receiving NAC-GS. However, the development of grade 3/4 AEs during AC did not have a similar impact on patients in the up-front surgery group. In addition, the development of grade 3/4 AEs during NAC-GS was also significantly associated with the development of grade 3/4 AEs during adjuvant chemotherapy. Our analyses of survival outcomes showed that among patients undergoing up-front surgery, the five-year survival rates in the patients developing grade 0/1/2 during AC (*n* = 41), patients developing grade 3/4 during AC (*n* = 13), and patients who did not receive AC (*n* = 17) were 25.3%, 20.5%, and 6.7%, respectively, with a borderline significant difference between the group that developed grade 0/1/2 AEs during AC and the no-AC group. On the other hand, in patients who underwent surgery after receiving NAC-GS, the five-year survival rates in the patients developing grade 0/1/2 during AC (*n* = 50), patients developing grade 3/4 AEs during AC (*n* = 15), and patients who did not receive AC (*n* = 11) were 33.0%, 0%, and 0%, respectively, with significant differences between the groups developing grade 0/1/2 AEs and grade 3/4 AEs during AC, as well as between the group that developed grade 0/1/2 AEs during AC and the no-AC group. A multivariate analysis identified the development of grade 3/4 AEs during NAC-GS, the use of a gemcitabine-based AC regimen, a serum albumin level of <3.5 g/dL, and an estimated glomerular filtration rate of <90 mL/min/1.73 m^2^ prior to the initiation of AC as predictors of the development of grade 3/4 AEs during AC [39]. Postoperative complications (Clavien-Dindo grade III or more) were encountered in 26.8% of the patients, but major postoperative complications were not associated with adverse events during AC. In this regard, careful management of adjuvant chemotherapy to maintain an adequate relative dose intensity is needed, especially in patients who develop significant AEs during NAC-GS.

Another previous study performed with the same patient cohort showed that adjuvant chemotherapy using gemcitabine monotherapy or S-1 monotherapy was less effective in patients who had shown an Evans Grade I response to NAC-GS as compared with those who showed an Evans Grade II/III response [32]. Among 79 patients who received NAC-GS prior to surgery, the response to NAC-GS was Evans Grade I in 20 patients (25.3%), Grade IIa in 46 patients (58.2%), Grade IIb in 11 patients (13.9%), and Grade III in 2 patients (2.6%). Of the 79 patients, 65 (82.3%) received gemcitabine monotherapy or S-1 monotherapy as AC. In patients whose pathological response to NAC-GS was Evans Grade I, the completion rates of NAC and AC were low and the relative dose intensity of AC was also low. The median survival time (MST) of patients who received up-front surgery followed by AC was significantly longer than that of the patients who received surgery alone without AC (23.7 months vs. 8.9 months, *p* = 0.004). Similarly, among patients who showed an Evans Grade IIa response to NAC-GS prior to surgery, the MST was significantly longer in patients who received AC after surgery than in those who did not (24.0 months vs. 10.2 months, *p* = 0.018). On the other hand, no such difference between patients who did or did not receive AC was observed among patients who showed an Evans Grade I response to NAC-GS prior to surgery (MST: 13.6 months vs. 6.7 months, *p* = 0.531). A multivariate analysis revealed that an Evans Grade II/III response to NAC-GS and a relative dose intensity during AC of ≥80% were associated with improved patient survival outcomes. Our data suggested that patients showing an Evans Grade I response to NAC-GS could be considered to be poor responders, and that therefore, other adjuvant regimes or other adjuvant treatment strategies without chemotherapy may need to be considered in these patients. In such situations, modified FOLFIRINOX or gemcitabine plus nab-paclitaxel regimens may be alternative options for AC.

## 5. Discussion

Our results revealed no survival benefit of NAC-GS in patients with resectable disease. In addition, they also showed that the resection rate and R0 resection rate were similar in patients who did and did not receive NAC-GS, although the rate of administration of AC was increased in the former group. Our results were in line with those of a previous meta-analysis in some respects [8] but different from those of the Prep-02/JSAP05 study, which showed a survival benefit of NAC-GS [25]. A recent meta-analysis also suggested that NAT is associated with a lower risk of development of lymph node metastasis, an improved R0 resection rate, and improved recurrence-free and overall survivals in patients with resectable disease [11]. Our negative results may be ascribed to the small number of patients, the different NAC regimens used, and the short duration of NAT. On the other hand, the pathological response rates to NAC-GS in our series were similar with those in a previous report [31] (Evans Grade I: 23.5% vs. 23%; Evans Grade IIa: 60.5% vs. 57%; Evans Grade IIb: 13.6% vs. 20%). The response rate to NAT may be different based on the regimen, duration, and additional use of radiotherapy. We have summarized the results of neoadjuvant studies and our series in Table 3 [31,32,40,41,42,43,44].

Our results also suggested that more effective regimens may be required for patients with BR-PV or BR-A disease. In this regard, gemcitabine plus nab-paclitaxel or modified FOLFIRINOX may be suitable alternative regimens. In addition, NACRT may be beneficial for patients with BR-A disease to secure negative surgical margins. Our results also showed that severe complications associated with NAC may affect the subsequent course of the multidisciplinary approach, and that completion of AC while maintaining a sufficient relative treatment intensity may be of importance to obtain longer patient survival periods. As mentioned above, intensive regimens, including triplet regimens such as FOLFIRINOX, have been widely introduced in the field of NAT; however, these regimens may not be applicable to Asian patients because of their considerable toxicity. Careful and thorough planning of the treatment strategy and meticulous management of NAT, surgery, and AT are required on a patient-by-patient basis, and the strategies required may be different between Eastern and Western countries.

Our previous results suggested clinical importance of the response to NAC. A favorable pathological response to NAC was linked with favorable patient outcomes after surgery [38]. The relationship between the pathological response to NAC-GS and survival outcome suggested that the tumor biological behavior could be assessed by the pathological response to NAC-GS, and that down-staging could be obtained by a good pathological response to NAC. In contrast, poor response, as well as the development of severe adverse events during NAC, were associated with severe adverse events during AC, a reduced dose intensity, and unfavorable outcomes. Our previous study also indicated that efforts to maintain a relative dose intensity of ≥80% and/or to complete the planned AC are of crucial importance [39]. Once again, careful management of adjuvant chemotherapy, including dose reduction where needed, are recommended based on observations during NAC. In this regard, short-term NAC may be beneficial as a tolerance test for the total management course in patients with PDAC. The significance of NAC as a predictor of the success rate of the total treatment has also been emphasized in previous systematic reviews and meta-analyses [8,9,10,11,27], in line with the suggestions based upon our results.

Immunotherapy has been shown to be less effective for PDAC as compared with other malignancies, and only a small subset of patients (mismatch repair-deficient or microsatellite–unstable PDAC) responds to immunotherapies. The poor response to immunotherapy may be ascribed to (1) intertumoral and intratumoral heterogeneity, (2) the composition of the tumor stroma, and (3) crosstalk with cancer cells [45]. Relevant results have shown that immunotherapy or even immunotherapy combined with standard chemotherapy is not effective for patients with advanced PDAC; however, the use of immunotherapy in an adjuvant or neoadjuvant setting may be worth considering in patients with a minimal volume of residual tumor cells and/or maintained immunological function. So far, the results of clinical trials of NACRT plus pembrolizumab, and GVAX vaccine combined with a PD-1 antagonist and CD137 agonist antibodies have been published [46,47]. More refinements of the combination regimens using immunotherapy and chemo(radio)therapy could provide a breakthrough novel treatment for PDAC.

## 6. Conclusions

We have summarized the problems associated with NAT and AT for patients with PDAC, partly based on our institutional experience. Accumulating evidence has revealed the benefits, impact, and adverse effects of NAT in recent years. Wide adoption of NAT will change the overall treatment strategies for PDAC. The results of ongoing clinical trials and further studies in the near future are required to analyze the appropriate balance between NAT and AT for patients with PDAC.

## Figures and Tables

**Figure 1 cancers-16-00910-f001:**
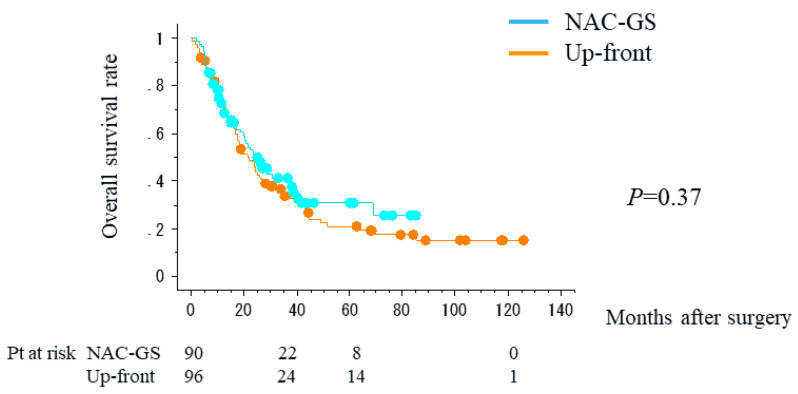
Overall survival curves of patients who received NAC-GS prior to surgery (NAC-GS) and patients who received up-front surgery (Up-front).

**Figure 2 cancers-16-00910-f002:**
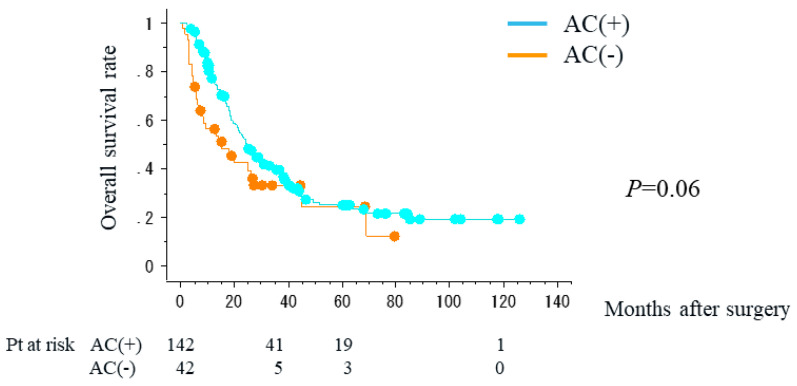
Overall survival curves of patients who received (AC (+)) but did not receive (AC (−)) adjuvant chemotherapy.

**Table 1 cancers-16-00910-t001:** Impact of NAC-GS therapy (*n* = 95).

Variable	Pre-NAC-GS (*n* = 95)	Post-NAC-GS (*n* = 95)	*p* Value	Post/Pre Ratio
Tumor size (mm)	24.6 (2.4–70.0)	18.2 (1.4–64.0)	<0.0001	0.82 (0.40–1.90)
CEA (ng/mL)	3.0 (0.8–163.0)	3.4 (0.7–97.5)	0.72	1.02 (0.25–4.92)
CA19-9 (U/mL)	203.0 (2–12,000)	84.5 (2–12,000)	<0.0001	0.63 (0.01–55.0)
DUPAN-2 (U/mL)	220.0 (25–14,000)	97.0 (4–3400)	<0.0001	0.66 (0.03–10.8)
Span-1 (U/mL)	52.0 (1–1600)	25.0 (1–1100)	<0.0001	0.68 (0.01–9.57)
Elastase-1 (ng/dL)	218.0 (25–8620)	150.0 (40–3613)	<0.0001	0.61 (0.01–8.17)

Data expressed as median (range). Wilcoxon signed rank test.

**Table 2 cancers-16-00910-t002:** Comparison between the NAC-GS (*n* = 95) and up-front surgery (*n* = 104) groups, stratified by resectable (R), borderline resectable-PV (BR-PV), and borderline resectable-A (BR-A) classifications. (+) = Yes; (−) = No.

	R Up-Front (*n* = 77)	R NAC-GS (*n* = 65)	*p* Value	BR-PV Up-Front (*n* = 20)	BR-PV NAC-GS (*n* = 16)	*p* Value	BR-A Up-Front (*n* = 7)	BR-A NAC-GS (*n* = 14)	*p* Value
Resection			0.14			0.16			0.99
(+)	72	64		18	15		5	10	
(−)	5	1		2	1		2	4	
R0 resection			0.60			0.03			0.67
Yes	60	48		13	14		2	5	
No	7	4		5	0		2	3	
Adjuvant chemotherapy			0.01			0.34			0.69
(+)	49	55		15	13		3	7	
(−)	22	9		4	2		2	3	

**Table 3 cancers-16-00910-t003:** Results of neoadjuvant studies and our series.

Author	Country	Period	Study Design	Number of Patients	NAT Regimen	Duration	Pathological Response	Median OS (Months)
Motoi et al. (2013) [31]	Japan	2008–2010	Prospective phase 2	35	Gemcitabine plus S-1	2 cycles (6 weeks)	Evans Grade I: 23%; Grade IIa: 57%; Garde IIb: 20%	19.7
Casadei et al. (2015) [40]	Italy	2007–2014	RCT	38	Gemcitaine plus radiation	12 weeks	Rebekah Grade Minimal: 11.1%; Small: 16.7%; Moderate: 27.8%; Large: 5.6%	22.4
Golcher et al. (2015) [41]	Germany	2003–2009	RCT	33	Gemcitabine plus cisplatin plus radiatioon	30 days	NR	17.4
Reni et al. (2018) [42]	Italy	2010–2015	RCT	29	Cisplatin plus epirubicin plus gemcitabine	3 months	Marked: 36%; Moderate: 32%; Poor: 32%	38.2
Versteijne et al. (2022) [43]	Netherlands	2013–2017	RCT	91	Gemcitabine plus radiation	10 weeks	NR	15.7
Seufferlein et al. (2023) [44]	Germany	2015–2019	RCT	59	Gemcitabine plus nab-paclitaxel	2 cycles (8 weeks)	Complete: 6.2%; Moderate: 3.1%; Minimal: 40.6%; Poor: 50.0%	25.5
Our series [32]	Japan	2013–2019	Retrospective	95	Gemcitabine plus S-1	2 cycles	Evans Grade I: 23.5%, Grade IIa: 60.5%; Grade IIb: 13.6%; Grade III: 2.4%.	22.0

NAT: Neoadjuvant Treatment; NR: Not reported; OS: Overall Survival; RCT: Randomized Controlled Trial; RFS: Recurrence Free Survival.

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
