# Peer review of "Neoadjuvant and Adjuvant Chemotherapy for Pancreatic Adenocarcinoma: Literature Review and Our Experience of NAC-GS"

_cancers, 2024, doi:10.3390/cancers16050910_

Round 1
Reviewer 1 Report
Comments and Suggestions for Authors
The authors present their experience with neoadjuvant Gem-S1 and summarize some emerging data in the use of neoadjuvant chemo for PDAC. Overall the paper is interesting. There are several issues to be addressed
1) Title- the title seems to be a little discordant with the data presented as the data are mainly on the use of neoadj Gem-S1
2) Abstract- nowhere in the abstract is evident that authors present their own data besides writing a review
3) Tables. Table 1- you need to give the n value for each column as in table 2
4) Results>line 161 What do the authors mean by GI stricture?
Give rationale/supporting data of using neoadj Gem/S1
line 183- how you assess tumor size?
line 188- give literature on markers besides the CA 19-9 as in most institutions only CA 19-9 is checked
language needs editing numerous mistakes
p values are missing in some areas of the results such as line 246-249
Discussion- 312-313: comment on the different path response rates of this study versus major RCTs on neoadjuvant
make a table that summarizes the results of important neoadjuvant RCTs and how they compared with the current study ie duration, agents, path response, RFS, OS
Comments on the Quality of English LanguageEditing is needed as there are numerous errors some examples
line 151 resectable 152 "power of NAT" there is no such thing as power of chemotherapy lines 213 and 214 spelling of year etc etc
Reviewer 2 Report
Comments and Suggestions for Authors
This is an exciting paper addressing a topic of interest. However, a few concerns should be addressed before acceptance:
The paper is presented as a review. However, it includes the authors' experience and analyses. I suggest submitting the manuscript in the format of an Original Article.
The paper analyses both neoadjuvant and adjuvant chemotherapies for resected PDAC. Please consider changing the title to reflect the content of the manuscript better. Furthermore, the same suggestion is for the Abstract.
Please consider referencing the statement: “In this regard, neoadjuvant therapies (NATs) are reasonable because NATs can be applicable for more patients scheduling curative resections with more reliable intensity of treatments.”
The statement “Herein, we discussed the remaining issues associated with NATs/ATs for PDAC and also reviewed our experience of neoadjuvant chemotherapy (NAC) using gemcitabine plus S-1.” is confusing. What does “remaining issues” mean?
Why did the patients with digestive stenosis were excluded from the study?
No data about complication rates or their impact on the adjuvant treatment are provided.
It is unclear how many patients received NAT and how many patients received up-front surgery.
A few comparative analyses are confusing.
A native English speaker must check the manuscript to improve fluency and correct a few minor errors.
Comments on the Quality of English LanguageModerate editing of English language required
Round 2
Reviewer 1 Report
Comments and Suggestions for Authors
appreciate the responses
Comments on the Quality of English Languageappreciate the responses
Reviewer 2 Report
Comments and Suggestions for Authors
The authors correctly addressed all major concerns raised by the reviewers
Comments on the Quality of English LanguageMinor editing of English language required